# Neurological Surveillance in Moderate-Late Preterm Infants—Results from a Dutch–Canadian Survey

**DOI:** 10.3390/children9060846

**Published:** 2022-06-08

**Authors:** Martine F. Krüse-Ruijter, Vivian Boswinkel, Anna Consoli, Ingrid M. Nijholt, Martijn F. Boomsma, Linda S. de Vries, Gerda van Wezel-Meijler, Lara M. Leijser

**Affiliations:** 1Department of Neonatology, Isala Women and Children’s Hospital, 8025 AB Zwolle, The Netherlands; martineruijter@hotmail.com (M.F.K.-R.); v.boswinkel@isala.nl (V.B.); g.meijler@isala.nl (G.v.W.-M.); 2Section of Neonatology, Department of Pediatrics, Cumming School of Medicine, University of Calgary, Calgary, AB T2N 4N1, Canada; aconsoli@ucalgary.ca; 3Department of Innovation & Science, Isala Hospital, 8025 AB Zwolle, The Netherlands; i.m.nijholt@isala.nl; 4Department of Radiology, Isala Hospital, 8025 AB Zwolle, The Netherlands; m.f.boomsma@isala.nl; 5Department of Neonatology, University Medical Center, 3584 EA Utrecht, The Netherlands; l.s.devries-23@umcutrecht.nl

**Keywords:** preterm infants, neuroimaging, neurological surveillance, follow-up, cranial ultrasound

## Abstract

Preterm birth remains an important cause of abnormal neurodevelopment. While the majority of preterm infants are born moderate-late preterm (MLPT; 32–36 weeks), international and national recommendations on neurological surveillance in this population are lacking. We conducted an observational quantitative survey among Dutch and Canadian neonatal level I–III centres (June 2020–August 2021) to gain insight into local clinical practices on neurological surveillance in MLPT infants. All centres caring for MLPT infants designated one paediatrician/neonatologist to complete the survey. A total of 85 out of 174 (49%) qualifying neonatal centres completed the survey (60 level I–II and 25 level III centres). Admission of MLPT infants was based on infant-related criteria in 78/85 (92%) centres. Cranial ultrasonography to screen the infant’s brain for abnormalities was routinely performed in 16/85 (19%) centres, while only on indication in 39/85 (46%). In 57/85 (67%) centres, neurological examination was performed at least once during admission. Of 85 centres, 51 (60%) followed the infants’ development post-discharge, with follow-up duration ranging from 1–52 months of age. The survey showed a wide variety in neurological surveillance in MLPT infants among Dutch and Canadian neonatal centres. Given the risk for short-term morbidity and long-term neurodevelopmental disabilities, future studies are required to investigate best practices for in-hospital care and follow-up of MLPT infants.

## 1. Introduction

Preterm birth (before 37 weeks of gestation) is associated with neonatal morbidity and lifelong health disabilities, including physical disabilities and neurodevelopmental and behavioural problems [1]. The vast majority (>80%) of the preterm population is born moderate to late preterm (MLPT) at a gestational age (GA) of 32–36 weeks [2]. Compared to full-term infants, MLPT infants are at increased risk for short-term morbidities such as respiratory distress, hypoglycaemia, hyperbilirubinemia, feeding difficulties, and sepsis [3,4]. Not all MLPT infants have the same risk of adverse events [5,6,7].

Despite the known adverse short-term outcomes, dedicated neurological surveillance for MLPT infants, such as neuroimaging and follow-up, has generally not been included in international and national accepted guidelines. An explanation for this might be that the needs of MLPT infants and their risks for long-term adverse outcomes were considered comparable to those of full-term infants (GA ≥ 37 weeks) [8]. Consequently, from a neurological perspective and after discharge, MLPT infants are often managed in the same way as their full-term counterparts.

Several studies have highlighted that MLPT infants have a higher risk for neurodevelopmental problems than previously considered. Compared to full-term infants, MLPT infants have an up to four-fold higher risk of cognitive, behavioural, and motor problems during childhood, which can persist into adolescence [9,10,11,12,13]. Immaturity of the brain and brain injury may contribute to these problems, but this association has yet to be thoroughly studied [14]. In an ongoing multicentre Dutch–Canadian study, we are investigating the relationship between neonatal neuroimaging findings and neurodevelopmental outcome in MLPT infants (Registered at: https://www.trialregister.nl/trial/6310, accessed on 17 May 2022).

In 2015, the Dutch Paediatric Association recommended performing neuroimaging in MLPT infants on indication, such as pregnancy complications, conditions associated with increased risk of brain abnormalities or neurological symptoms [15]. In addition, a few recommendations for neuroimaging and follow-up in MLPT infants have recently been published. In 2020, the American Academy of Paediatrics and the Canadian Paediatric Society advised to perform cranial ultrasound (cUS) in infants born between 32 + 0–36 + 6 weeks’ gestation with risk factors for brain injury (e.g., placental abruption, prolonged mechanical ventilation, sepsis, major surgery, or abnormal neurological symptoms) [16,17]. The Spanish Society of Neonatology, in collaboration with the Spanish Association of Primary Paediatric Care, recommended post-discharge neurodevelopmental follow-up in all preterm infants or at least all infants with risk factors for adverse neurodevelopmental outcomes [18]. Most of the recommendations were based on expert opinions, as qualitative research is currently lacking. The Dutch and Canadian Paediatric Societies do not provide recommendations on follow-up in MLPT infants [19,20].

Considering the growing evidence of short- and long-term problems of MLPT infants, more accurate neurological surveillance and follow-up is required. The above-mentioned recommendations are relatively recent, mostly based on expert opinion and panel consensus. Implementation in centres in the Netherlands and Canada is expected to show great variation.

The aim of this study was to gain more insight into local clinical practices for MLPT infants, with a focus on neurological surveillance including neuroimaging and standardized follow-up, by conducting a survey among paediatricians and neonatologists in the Netherlands and Canada. A secondary aim was to compare the clinical care practices between these two countries. Better insight in practice differences may lead to clinical practice changes and improved care.

## 2. Materials and Methods

The current study was an initiative of research teams at the Isala Women and children’s Hospital (Zwolle, The Netherlands) and the Calgary Zone Neonatal program (Calgary, Canada). The survey (see Appendix A) was conducted in three steps between June 2020 and August 2021.

In step 1, all Dutch and Canadian centres providing care for MLPT infants (GA 32 + 0 to 36 + 6 weeks) were identified. In the Netherlands, this applied to 69 centres, of which 10 had a level III Neonatal Intensive Care Unit (NICU). In Canada, this applied to 105 centres, of which 31 centres had a level III NICU. In step 2, all qualifying centres were approached and asked to designate one paediatrician/neonatologist to complete the survey, indicating a preference for a clinician with the most exposure or local responsibility for the care of MLPT infants. In step 3, an e-mail was sent to these designated paediatricians/neonatologists, asking them to complete the survey. The email included a personal and direct link to the survey. All answers/ data were anonymized.

Before distribution of the survey, the survey was tested within our own research teams, including members with and without medical background, and revised based on the feedback. Subsequently, we additionally sent the survey to three paediatricians/neonatologists from different centers for validation.

The first part of the survey consisted of general questions such as number of (incubator) beds in their unit, profession of the responder (paediatrician, neonatologist), and number of years of working experience as staff. The second part of the survey, focused on the clinical care of MLPT infants, consisted of items concerning admission criteria, routine laboratory testing, and neurological surveillance, including neuroimaging (cUS and magnetic resonance imaging (MRI)), neurological examination during admission, and follow-up after discharge. In addition, we asked questions about management of MLPT infants admitted to the maternity ward and participants’ opinion on current overall management of MLPT infants in their unit (see Appendix A). All questions were multiple choice, with free text options to specify the answer and comments. Once the survey was completed, it could not be accessed again. All answers were stored automatically in Research Manager (version 5.56.0, Research Manager, Deventer, The Netherlands). The study was approved by the Medical Ethics Committee at the Isala Hospital Zwolle (Ethics ID 200423) and the University of Calgary Conjoint Health Research Ethics Board (Ethics ID REB20-0442; 9 June 2020). Data were analysed using SPSS software (version 26.0; SPSS inc, Armonk, NY, USA). Frequencies are reported as number and percentage. We used descriptive approaches to present the survey answers from participating Dutch and Canadian centers as our study was aimed at exploring variations between Neonatal centers rather than a direct comparison between Dutch and Canadian centers.

## 3. Results

Of the 174 identified centres, 75% centres designated a paediatrician or neonatologist to complete the survey. In total, 49% of the identified centres completed the survey (Figure 1), of whom 54% were Dutch and 46% were Canadian. Characteristics of participants and their neonatal centres (e.g., level II or III, number of incubator beds) are shown in Table 1.

### 3.1. Admission Criteria

In 92% of participating centres, admission of MLPT infants was based on infant-related criteria. Seventy-one percent of centres used GA as an admission criterion. GA below 35 weeks (28%) and below 36 weeks (32%) were the most frequently reported admission criteria. In addition, 58% of centres used birth weight (BW) as a criterion for admission. In total, 3 centres (4%) determined the need for admission based on BW percentile (below <P5 or <P10), while 46 centres (54%) used an absolute weight as cut-off for admission, ranging from <1800 to <2500 g. The most frequently reported cut-off for admission was a BW < 2000 g (20%).

Furthermore, clinical practices included admission criteria for hypoglycaemia for postnatal days 1–3 (61%; cut-off value range: 1.6–3.3 mmol/L), hyperbilirubinemia needing treatment (65%), (suspected) sepsis (65%), feeding difficulties (57%), suboptimal start indicated by an APGAR score < 7 at five minutes (35%), and respiratory distress (78%). Other admission criteria mentioned by participants were maternal medication use (not specified), maternal diseases that may affect the infant’s condition (e.g., hypothyroidism, gestational diabetes), neonatal abstinence syndrome, congenital anomalies (e.g., congenital heart disease), and (suspected) seizures. Most centres (86%) recorded their practices in local guidelines.

### 3.2. Laboratory Testing

Out of 85 centres, 46 (54%) indicated performing routine blood sample testing in MLPT infants. Testing included concentrations of glucose (routine: 45%; only when indicated (i.e., risk factors or clinical symptoms): 9%), bilirubin (routine: 32%; only when indicated: 20%), C-reactive protein (routine: 1%; only when indicated: 38%), and haemoglobin (routine: 22%; only when indicated: 22%). Other reported routine blood testing included: ferritin measurement two weeks after birth (1%), complete blood count in small for GA infants (2%), and electrolytes (4%).

### 3.3. Neuroimaging

Local guidelines regarding neuroimaging (cUS/MRI) in MLPT infants were available in 49% of the centres.

CUS: Sixty-five percent (55/85) of the centres indicated to perform cUS in MLPT infants. In 16/85 (19%) centres, cUS was performed routinely in (a subgroup of) MLPT infants. GA cut-off for routine cUS differed between centres, ranging from <33 weeks in 38% of centres, <34 weeks in 31%, <35 weeks in 13%, and no GA cut-off in 19%. In the centres with routine cUS, cUS was performed on day 1–3 (31%), on day 4–7 (38%), or just prior to discharge, or around term equivalent age (6%). In the remaining 25% of centres, the day when cUS was performed was not reported. CUS was repeated based on findings on the first cUS in 56% of centres and in 6% cUS was routinely performed twice during admission; for 19% of the centres this question was not completed. In the remaining 39 out of the 55 centres (46%), cUS was not routinely performed based on preterm gestation but rather on other indications such as meningitis (Table 2).

Brain MRI: In none of the neonatal centers, MRI was performed routinely. Twelve percent (*n* = 14) of the centres indicated performing MRI in MLPT infants on indication, including abnormal cUS findings, hypoxic-ischemic encephalopathy, or clinical signs that can be associated with brain abnormalities (e.g., seizures, meningitis, microcephaly, suspected anomalies, or dysmorphic features). The remaining 70 centres did not perform MRI or referred MLPT infants to a level III neonatal centre for MRI (the latter was only mentioned by five centres).

### 3.4. Neurological Examination

Routine neurological examination in admitted MLPT infants was performed in 67% of the centres. In total, 20% of the centres performed neurological examination on the first postnatal day and 21% prior to discharge, while 26% performed examinations on both the first postnatal day and prior to discharge, or even more frequently. The content of the neurological examination was not specified.

### 3.5. Follow-Up Program

Follow-up of MLPT infants was performed in 60% of the centres. In almost all centres, the infants were seen by a paediatrician or neonatologist (96%), often together with other disciplines (e.g., physiotherapists) or allied health services (e.g., vaccinations or repeating cUS examination) (80%) (Table 3). Duration of follow-up ranged from 1 to 52 months after birth.

### 3.6. Care for MLPT Infants Admitted to the Postpartum Maternity Ward

Questions about the care of MLPT infants staying in the postpartum maternity ward were interpreted differently. Seven participants did not complete any of the questions, 15 used the options ‘I don’t know’ or ‘I don’t wish to answer this question’, and two completed the questions but commented that they did not fully understand the questions. Hence, we cannot elaborate on these answers

### 3.7. Differences between Dutch and Canadian Practices

Among the Dutch centres, 6 out of 10 (60%) eligible level III centres and 40 out of 59 (68%) eligible level I–II centres completed the survey. Among Canadian centres, 20 out of 31 (65%) eligible level III centres and 19 out of 74 (26%) eligible level I–II centres completed the survey.

Almost all Dutch and Canadian neonatal centres used criteria for routine admission of MLPT infants, which were recorded in local guidelines (Dutch, 92%; Canadian, 92%). Of these, 81% of the Dutch centres used both GA and birthweight as admission criteria versus 65% of the Canadian centres. Routine laboratory testing was performed in 54% Dutch centres and 54% Canadian centres. Of the 46 Dutch centres, 74% had local guidelines for neuroimaging, compared to only 21% of the Canadian centres. CUS examinations were performed in 70% of the Dutch and in 59% of the Canadian centres. Routine cUS (in a subgroup of MLPT infants) was performed in 22% Dutch and 13% Canadian centres. MRI was not routinely performed in either Dutch or Canadian responding centres. Two Dutch (4%) and eight Canadian (21%) centres performed MRI in MLPT infants on clinical indication. Routine neurological examinations were performed in 61% Dutch and 74% Canadian centres. In Dutch centres, 80% provided some form of follow-up for MLPT infants, while this was the case in 36% Canadian centres. Five Canadian centres (13%) without follow-up commented that follow-up was provided by other health care providers (i.e., general practitioners or by a local community paediatrician).

## 4. Discussion

We conducted an online survey to gain insight into the current clinical care practices on neurological surveillance of MLPT infants in neonatal centres across the Netherlands and Canada. The survey results highlight the variability in clinical care provided to MLPT infants among neonatal centres and between Dutch and Canadian centres, during the neonatal period as well as beyond.

Different GA and BW criteria for admission of MLPT infants to the neonatal unit were reported among participating centres. Our findings are consistent with those from a national survey among 180 neonatal units in England by Fleming et al. [21]. Using a short survey, consisting of five questions related to GA and BW criteria for admission of late preterm infants (GA 34 + 0–36 + 6 weeks), they reported variability in the GA and BW admission criteria. The GA limit for admission ranged from 34 to 37 weeks and the BW limit ranged from 1500 to 2500 g. The variation in admission criteria is likely related to differences in local facilities, organization of health care provision, and skills level of personnel in the different units (neonatal units or postpartum maternity ward), as well as lack of evidence for best practices in MLPT infants.

We acknowledge that there are other health care professionals, such as primary care/community paediatricians, that provide care and follow-up in MLPT infants, which may explain some of the differences between Dutch and Canadian centres.

Nearly half of the centres participating in our survey did not perform routine blood testing in admitted MLPT infants. Both the Dutch and Canadian Paediatric Society guidelines advise to check blood glucose levels within 2 h of birth in all preterm infants (GA < 37 weeks) to allow for early detection of hypoglycaemia and therewith prevention of potential brain injury [22,23]. Kerstjens et al. reported that neonatal hypoglycaemia is associated with an increased risk of neurodevelopmental delay in MLPT infants at 4 years of age [24]. As MLPT infants may thus benefit from more strict monitoring and management of blood glucose levels, implementation in routine clinical practice is recommended.

Some participants reported routine testing of other blood parameters (such as electrolytes, haemoglobin, and C-reactive protein). To our knowledge, evidence supporting this practice is lacking. More specific, routine testing of C-reactive protein has been shown to have low sensitivity and specificity, and positive predictive value for neonatal sepsis [25,26]. For some blood parameters, routine testing is not necessary. Although all preterm infants (GA < 37 weeks) have an increased risk for hyperbilirubinemia, current Dutch and Canadian guidelines on the management of hyperbilirubinemia in newborn infants propose a systematic risk assessment rather than routine monitoring of total serum bilirubin for infants GA ≥ 35 weeks [27,28]. In addition, a growing number of centres use transcutaneous bilirubin concentration measurements for risk assessment and to reduce the number of unfavorable pokes for infants required to measure total serum bilirubin. Further studies are needed to evaluate whether other routine laboratory tests, in addition to glucose monitoring, may be beneficial in MLPT infants.

Only 21% of the Canadian centres had a local guideline for neuroimaging in MLPT infants, while 74% of Dutch centres had such a guideline. The recommendation from the Canadian Pediatric Society’s on neuroimaging of the preterm infant’s brain is still recent and may not yet be embedded in local Canadian guidelines while the Dutch recommendations on neuroimaging in preterm infants date from 2015 [15,16]. Local differences in care organization, with neonatal teams mostly performing the cUS in Dutch centres and radiology teams in Canadian centres, may have contributed to the above difference in neuroimaging. Of note, cUS was performed routinely in MLPT infants in ten Dutch and five Canadian centres, while this is not recommended by current Dutch, American or Canadian reports [15,16,17]. Furthermore, although American and Canadian recommendations advise cUS in MLPT infants with risk factors for brain injury, the level of evidence for this recommendation was moderate (grade B) in a recent review [29]. The above stresses the need for more research on the risk factors for brain injury in MLPT infants and potential subsequent neurodevelopmental problems. A better understanding may help to target neuroimaging in a subgroup of MLPT infants most at risk for brain abnormalities and avoid unnecessary diagnostic imaging in infants at low risk. As MLPT infants comprise over 80% of all preterm infants, unnecessary diagnostic procedures would mean an enormous burden on health care and the available budget. 

Our survey also demonstrated large differences in performing neurological examinations and follow-up in MLPT infants. A surprisingly large number of centers do not routinely perform neurological examinations. A routine and well executed neurological examination by an experienced examiner (e.g., according to Amiel-Tison or Dubowitz or the HINE neurological examination) may benefit MLPT infants. The exam may contribute to identifying the infants at risk for problems, in particular abnormal motor development, that would benefit from closer follow-up and/or early intervention. Unfortunately, we did not ask participants for details on the physical and neurological examination performed in their centers. The lacking information restricts us to elaborate on why routine neurological examination was not performed in 33% of centres. Furthermore, this leaves the possibility that a neurological examination was captured in the routine physical examination and the numbers of neurological exams performed would be higher.

General movements assessments (GMA) have been shown to have predictive value for motor outcomes, in particular cerebral palsy, in an early stage [30]. Implementation of routine neurological examination and post-discharge follow-up, in combination with GMAs around term equivalent age and post-term, can, therefore, be beneficial in MLPT infants. Early identification of developmental delay, or risk thereof, is important to facilitate timely intervention and therewith optimize neurodevelopmental outcomes [3,31].

So far, a few ‘expert opinion based’ recommendations for the follow-up of the vulnerable MLPT population have been reported [18,32]. As the MLPT population is large, seeing all MLPT infants back for developmental assessments will place a considerable burden on health care services and again be of major cost to society [33,34]. Thus, there is a need for research on best practices and cost-effectiveness of follow-up for MLPT infants enabling targeted follow-up for MLPT infants at highest risk for abnormal development. 

Our study highlights the variability in practice on neurological surveillance of MLPT infants across neonatal centers caring for these infants. Furthermore, it stresses the need for more research in the MLPT population and a more uniform and targeted clinical practice. Ultimately, this may improve care for and outcomes of MLPT infants. We hope that our data, in combination with other literature stressing the increased risk of MLPT infants for later neurodevelopmental problems, will contribute to the implementation of a neurological examination in the routine care of MLPT infants.

Our study also has some limitations. Firstly, participation in the survey was voluntary. Consequently, clinicians with a special interest in neonatal neurology may have been selected. In addition, only 26% of the level I-II centres in Canada while 68% of those in the Netherlands were represented in this survey. This may have influenced the evaluation of differences in practices between the Netherlands and Canada. In 35% of centres, cUS was not performed in MLPT infants. Unfortunately, we did not ask for participants to specify the reason for not performing cUS. Therefore, we cannot comment on whether reasons included practical limitations or limited awareness of the risks/symptoms of brain injury in MLPT infants. In addition, as mentioned above, we did not specify the neurological examinations, nor did we add a question on GMA.

Finally, the response rate of our survey (49%) was moderate but comparable to those of other surveys on clinical practices in paediatrics [35,36]. The study was conducted in two high-income countries and is probably not representative for the global situation.

## 5. Conclusions

In conclusion, this study demonstrated a wide variety in neurological surveillance of MLPT infants. National guidelines advise monitoring of blood glucose concentrations for early detection of hypoglycaemia, but practices are not consistent across neonatal centres. There is a lack of evidence for best practice on admission criteria, neuroimaging and follow-up for MLPT infants. Future studies need to investigate whether subgroups of MLPT infants may benefit from advanced in-hospital care (such as neuroimaging and additional laboratory testing) and post-discharge neurodevelopmental follow-up.

We hope to stimulate a more uniform practice for MLPT infants between and within countries, thus enabling benchmarking and improvement of care, with the overall aim to improve outcomes of MLPT infants and their families. However, given the MLPT population size, being 80% of the preterm population, redundant use of advanced care should be avoided.

## Figures and Tables

**Figure 1 children-09-00846-f001:**
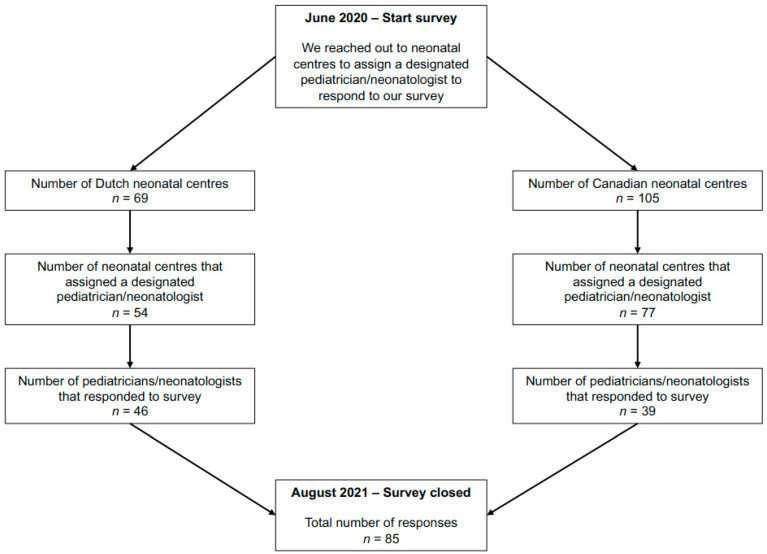
Recruitment of participants.

**Table 1 children-09-00846-t001:** Characteristics of participants and setting.

	Dutch Centres	Canadian Centres	Total
Response rate	46/69 (67)	39/105 (37)	85/174 (49)
	*n* = 46	*n* = 39	*n* = 85
Position			
Paediatrician	22 (48)	9 (23)	31 (36)
Neonatologist	23 (50)	29 (74)	52 (61)
Missing data	1 (2)	1 (3)	2 (2)
Number of years working as a paediatrician or neonatologist			
<5 years	5 (11)	4 (10)	9 (11)
5–10 years	13 (28)	10 (26)	23 (27)
10–15 years	14 (30)	6 (15)	20 (24)
>15 years	14 (30)	18 (46)	32 (38)
Missing data	0	1 (3)	1 (1)
Setting			
Level III neonatal centre	6 (13)	20(51)	25 (29)
Level I-II neonatal centre	40 (87)	19 (49)	60 (71)
Number of incubators/beds (in total)			
<5 beds	6 (13)	2 (5)	8 (9)
5–12 beds	12 (26)	6 (15)	18 (21)
10–15 beds	12 (26)	5 (13)	17 (20)
15–20 beds	13 (28)	7 (18)	20 (24)
>20 beds	3 (7)	19 (49)	22 (26)

Numbers are reported as number of centres and percentage, *n* (%).

**Table 2 children-09-00846-t002:** Indications for cUS in MLPT infants.

Indication	Number of Centres Performing cUS on Indication *n* = 39
(Suspected) seizures	36 (92)
Other neurological symptoms (such as jitteriness, irritability, excessive crying, abnormal muscle tone, lethargy)	36 (92)
Suspected sepsis	7 (18)
Confirmed sepsis	16 (41)
Suspected meningitis	22 (56)
Confirmed meningitis	32 (82)
Anaemia, infant needing PRBC transfusion	20 (51)
Hyperbilirubinemia, infant needing exchange transfusion	21 (54)
Antenatal diagnosis or suspicion of brain anomaly	37 (95)
Dysmorphisms	34 (87)
Other (multiple answers per centre)	15 (39) Intrauterine growth restriction/dysmaturity: 5 (13) BW < 1500 g: 3 (8) Unexplained apnea: 3 (8) Monochorionic twins: 2 (5) Resuscitation: 2 (5) Fetal therapy: 1 (3) Hydrocephalus: 1 (3) Micro-/macrocephaly: 1 (3) Severe hypoglycaemia: 1 (3) Severe thrombocytopenia: 1 (3) Suspected congenital cytomegalovirus: 1 (3)

Numbers are reported as *n* (%).

**Table 3 children-09-00846-t003:** Indication for follow-up, content, and duration of follow-up programs in MLPT infants.

Indication for Follow-Up *n* = 51		Follow-Up Performed by (Multiple Answers Were Possible) *n* = 51	
GA < 35 weeks	26 (51)	Paediatrician/-neonatologist	49 (96)
GA < 34 weeks	8 (16)	Paediatric nurse	7 (14)
GA < 33 weeks	9 (18)	Paediatric resident	3 (6)
GA < 37 weeks:	3 (6)		
GA < 36 weeks:	2 (4)		
Other	Low BW 38 (75) Medical conditions: 23 (45) Feeding difficulties: 2 (4) Psycho-social circumstances: 1 (2)	Paediatric nurse practitioner/ physician assistant	11 (22)

Content of follow-up visit *n* = 51		Collaboration with other disciplines or services (multiple answers were possible) *n* = 51	
Measuring weight and height	49 (96)	No	10 (20)
Physical examination	48 (94)	Yes	41 (80)
Neurological examination	44 (86)	Physiotherapist	30/41 (73)
Answering parents’ questions	48 (94)	Psychologist	7/41 (17)
Developmental assessment	46 (90)	Speech therapist	19/41 (46)
Other	Start iron supplementation: 2 (4) Screening for postnatal depression: 1 (2)	Other	Dietician: 5 (10) Well baby doctor or nurse: 3 (6) Social worker: 3 (6) Occupational therapist: 3 (6) Vaccinations: 3 (6)

Duration of follow-up program	*n* = 51		
1 month	1 (2)		
3–12 months	17 (33)		
When the child starts walking	12 (24)		
Unknown	2 (4)		
Other	19 (37) Up till 18 months: 1 (2) Up till 24 months: 4 (8) Between 24–36 months: 1 (2) Between 24–48 months: 1 (2) Up till 36 months: 2 (4) Up till 42 months: 2 (4) Up till 48 months: 2 (4) Up till 54 months: 1 (2) Depending on medical condition and history: 4 (8) First year in neonatal centre, after this the infant will enrol in a special program at the ‘well baby’ clinic: 1 (2)		

Numbers are reported as *n* (%).

## Data Availability

All the relevant data are within the manuscript.

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
