# Peer review of "Neurological Surveillance in Moderate-Late Preterm Infants—Results from a Dutch–Canadian Survey"

_children, 2022, doi:10.3390/children9060846_

Round 1
Reviewer 1 Report
The authors conducted an observational quantitative survey among Dutch and Canadian neonatal centres (June 2020 - August 2021) to collect information on local clinical practices on neurological surveillance in moderate and late preterm (MLPT) infants. Eighty- 22 five of 174 (49%) qualifying neonatal centres completed the survey. Admission of MLPT infants was based on clinical criteria in 78/85 (92%) centres. Cranial ultrasonography to screen the infant’s brain 24 for abnormalities was routinely performed in 16/85 (19%) centres and on indication in 39/85 25 (46%). In 57/85 (67%) centres, neurological examination was performed at least once during admission. Fifty-one of 85 (60%) centres followed the infants’ development post-discharge, with follow-up duration ranging from 1-52 months of age. The survey showed a wide variety in neurological surveillance in MLPT infants among Dutch and Canadian neonatal centres.
This is an interesting questionnaire study on practices in MLPT infants with focus on neurological examinations and follow-up.
Major concerns
While the study has its focus on neurodevelopment, I am missing general movement analysis. Particularly, as there is few data on this population and most studies look at VLBW infants. Please add at least some discussion on GMs.
Further, it is astonishing that many centers do not perform neurological examination. Please comment.
Minor comments
None
Reviewer 2 Report
This manuscript presents the results of a survey-based study of neonatal units in Canada and The Netherlands seeking to assess the current state of monitoring and care for moderate-late preterm infants. The investigators are well known in the field and have provided a generally well-written description of their study. There are several points, however, that the authors could consider to potentially improve the current version of the manuscript.
Main points include:
1. The authors do not provide any description of how the survey tool was tested and validated prior to widespread distribution. If not tested, they should provide an explanation for why that key step in survey development was not performed, especially given the concerns that respondents stated that some of them "did not fully understand the questions" (line 193).
2. The authors should consider the validity of directly comparing results between the two countries given that the makeup of the units in the different countries appeared quite different (more large, level III centers responded from Canada and more small, level I-II from The Netherlands) - this center difference is likely also the cause for there being more neonatologists responding from Canada and more non-neonatologist pediatricians in The Netherlands. If the authors keep this inter-country comparison, they should consider performing it in the context of the population differences (this could potentially include additional analyses to compare responses between different unit types and/or different provider types). It will be important to consider the goal of any comparisons being performed. Comparisons of level III vs level I-II or between different types of providers could potentially be used to target further studies or educational interventions (for instance to improve glucose monitoring or to decrease unnecessary procedures in this population). Given the discussion (lines 258-262), it seems that the countries are being compared to see if the different guidelines between the countries result in different practices; though if this is the case, it this should be directly stated at some point(s) in the manuscript.
3. The primary question regarding the use of cUS is not currently clear, and could use some further explanation. As currently described, it appears that ~35% of units would not perform cUS even if clinically indicated. It is unclear why any unit would select this option unless they don't have the ability to perform cUS in their hospital. Is this how the authors interpret those results? If so, please further discuss in the manuscript in light of the limitation that the survey did not specifically clarify this aspect of the question (i.e. why they would never perform a cUS). Since the authors focus quite a bit on the cUS question, this is a particularly important point to clarify.
4. Similarly, the interpretation of the "opinion on local management" has potential for significant selection bias given how the respondents were self-selected. This reviewer acknowledges that the authors mention this in their limitations; however, given this concern of selection bias specifically affects this section (it is the only section of the survey that is opinion-based and not general unit practice-based) and the authors did not find the results of this section important enough to mention in the discussion, this reviewer wonders about the value of keeping these results in the manuscript at all.
More minor points:
Abstract
- It is unclear why the authors use the parentheses in the word "international" (lines 17 and 46)
- Consider adding a mention of the different center types included in the study
- Lines 22-23: this sentence is a little confusing, centering around the terminology of "based on clinical criteria" which brings up the question of what the other option is (i.e. "non-clinical criteria"?). Consider restating as something more along the lines of "Clear criteria for MLPT admission were available for..." to better explain that the differentiation is between using criteria vs not using, as opposed to clinical criteria vs some other kind of criteria (if this reviewer is interpreting this correctly).
Intro
- Consider providing a citation for the last sentence of the first paragraph (line 43)
- Overall, this section is quite long. Consider condensing paragraphs 3-5 into a single summary paragraph, potentially with a table to summarize the inpatient vs. follow up recommendations between the different societies.
Methods
- A definition of "neurological examination" should be provided, assuming that it was provided to the respondents. Since a full physical examination should be performed at the very least once during each hospital stay, and a full examination should include at least a partial neurological exam, this reviewer would have expected that 100% (not 2/3) of units would have stated that they perform neuro exams on these patients. Do the authors feel these results represent a variable understanding of the question or a true finding that many of these units are not performing any aspects of the neurological exam on a regular basis.
- Was guidance provided for units with regards to who they should select in Step 2 to perform the survey (to get at the question of selection bias)
Results
- How was "Apgar <7" selected as the criteria for "suboptimal start"? Do the authors think that there may have been units that selected "no" to this question that did so because their definition of "suboptimal start" was different from the one stated?
- Line 168: this sentence should be reworded for clarity
Discussion
- Given the considerable overlap in scope between this current work and the Fleming study (reference 19, line 233), the authors should provide a more extensive discussion/comparison of the results of these two similar studies
- Is there guidance for bilirubin monitoring in Canada or The Netherlands that would be more applicable to discuss than the AAP guidelines? If so, consider adding and discussing those references.
- The end of the sentence on line 258-259 is missing a verb
- Consider revising the sentence on lines 286-287. This study does not necessarily highlight "knowledge gaps" since there is often a disconnect between "knowledge" available in the literature and practice. The survey results instead demonstrate variability in practice (which may or may not be due to knowledge gaps).
Abbreviations
- BIMP does not appear to be used anywhere in the manuscript text. It is mentioned in line 313, but since this is the only location, consider just spelling out in this single instance.
